# Factors affecting the prevention of unwanted pregnancies among young adolescents in secondary schools in the Eastern Province of Rwanda: An explorative qualitative study

**Thierry Claudien Uhawenimana** [ORCID]*, **Kellen Muganwa, Marie Chantal Uwimana, Marie Grace Sandra Musabwasoni, Olive Tengera, Joy Bahumura, Joella Mukashyaka, Jackline Mukakamanzi, Emmanuel Habyarimana, Innocent Ngerageze, Hellen Nwanko Chinwe, Emmerance Uwingabire, Francoise Mujawamariya, Richard Nsengiyumva, Oliva Bazirete**

School of Nursing and Midwifery, College of Medicine and Health Sciences, University of Rwanda, Kigali, Rwanda

* tcuhawenimana@gmail.com

## Abstract

### Introduction

The escalating number of teenage pregnancies, especially in the Eastern province of Rwanda, continues to raise concerns regarding the sexual and reproductive health of adolescents in the country. Recent statistics indicate that 5% of adolescent girls between the ages of 15 and 19 have given birth and 4% are currently pregnant with their first child. This highlights the critical need for comprehensive sexual and reproductive health education tailored for adolescents. However, there is limited evidence on factors affecting adolescents' efforts to prevent unwanted pregnancies and sexually transmitted infections in the Eastern Province of Rwanda, and the support systems available to adolescents in secondary schools, including the role of parents in promoting their sexual and reproductive health to minimize the risky sexual practices.

### Methods

An explorative qualitative study utilizing focus group discussions was conducted to garner the perspectives of 118 adolescents enrolled in six twelve-year-basic-education schools from three districts of the Eastern Province of Rwanda. Thematic analysis was employed to identify themes related to the impact of various factors on adolescents' sexual and reproductive health as they navigate through the physical and emotional changes from puberty to adolescence.

### Results

Adolescents are aware of the potential consequences of engaging in unprotected sexual intercourse which include the risk of unplanned pregnancy, sexually transmitted infections,

**Data Availability Statement:** All relevant data are within the manuscript and its Supporting Information files.

**Funding:** The author(s) received no specific funding for this work.

**Competing interests:** The authors have declared that no competing interests exist.

HIV/AIDS, and emotional distress. Female participants emphasized that young girls who do not receive adequate support upon becoming pregnant prematurely may encounter depression, discontinue their education, face the harsh reality of extreme poverty, and struggle significantly in assuming the responsibilities of parenthood as young single mothers. Adolescents highlighted the lack of parental guidance concerning sexual and reproductive matters as a significant obstacle in their pursuit of a healthy and safe sexual and reproductive health during adolescence.

## Conclusion

Inadequate parental engagement still hinders adolescents in navigating the physical bodily, mental, and emotional changes during adolescence. This affects their capacity to make well-informed decisions to prevent adverse consequences such as unintended pregnancies, substance misuse, sexually transmitted infections, and HIV/AIDS resulting from unsafe sexual practices. Since this study was qualitative, quantitative data necessary for a precise evaluation of the extent of the problem related to the absence of parental involvement in educating adolescents on sexual and reproductive health needs further research.

## Introduction

Adolescence is defined as a period spanning between 10 and 19 years of age [1]. This period is characterized by significant physical, emotional, and cognitive changes as adolescents navigate their way towards maturity [2, 3]. This transitional phase is often marked by a desire for independence, exploration, and self-discovery as adolescents begin to establish their own identities [3]. Although adolescence can present challenges for certain individuals who struggle to navigate the emotional changes that occur during this phase, it is also a crucial period of immense growth and development [3]. If effectively managed, it can serve as the building block for a prosperous and satisfying adulthood [3, 4]. During adolescence, some adolescents often start engaging in sexual relationships, a situation that, if not well managed, can lead to early and unintended pregnancies [5]. These pregnancies can occur due to various reasons such as lack of planning, failure to use contraceptives, sexual assault, and neglecting to use protection [6].

Thus, in order to promote the well-being of adolescents, it is essential to safeguard their sexual and reproductive health. In the context of adolescence, the World Health Organization underscores the importance of ensuring healthy adolescent sexual and reproductive health in terms of the physical and emotional wellbeing by ensuring adolescents' ability to remain free from unwanted pregnancy, unsafe abortion, sexually transmitted infections (STIs) including Human immunodeficiency virus/ Acquired immunodeficiency syndrome (HIV/AIDS), and all forms of sexual violence and coercion [7, 8]. For the purposes of this study, sexual and reproductive health further extends to adolescents' access to Sexual and Reproductive Health (SRH) care and satisfaction in meeting their needs to access to and utilization of contraception, information about SRH, prevention of STDs including HIV/AIDS, prevention of teen pregnancy.

Adolescents across the globe face numerous challenges that include sexual coercion and violence within intimate relationships, insufficient knowledge regarding sexuality and sexual and reproductive health, untimely and unintended pregnancies, restricted access to reproductive health services, especially contraception and safe abortion, gender inequalities, and

coerced marriages [9]. Additionally, adolescents' risk of contracting sexually transmitted infections (STIs), including HIV and viral hepatitis is high [9, 10]. For example, in 2022, an estimated 1.65 million adolescents were affected by HIV globally [11]. Research has also revealed that from 2010 to 2019, adolescents experienced the highest prevalence of sexually transmitted infections worldwide, with an incidence rate of 18,377.82 per 100,000 person-years (ranging from 14,040.38 to 23,443.31) [12]. Sub-Saharan Africa continues to record high rates of unintended teen pregnancies, especially among those aged 15–19 from low-income families residing in rural areas [13]. In 2019 alone, an estimated 21 million pregnancies occurred in this age group, with approximately half of them from Sub-Saharan Africa [13]. The situation is particularly dire for girls aged 10–14, with the global adolescent birth rate estimated at 1.5 per 1000 women in 2022 [13].

The statistics presented are concerning, highlighting the urgency to enhance efforts in addressing the issue of teenage pregnancies worldwide. In order to effectively mitigate the burden of teen pregnancies, it is crucial to conduct research that can identify the underlying factors contributing to the rise in unintended pregnancies. By doing so, interventions such as comprehensive sexual education for adolescents and the provision of accessible sexual and reproductive health services can be tailored to meet the specific needs and expectations of young individuals, ensuring that they are user-friendly and effective in reducing teenage pregnancies.

Target 3.7 of the sustainable development goals (SDGs) emphasizes the importance of universal access to sexual and reproductive health-care services by 2030 [14, 15]. This includes not only family planning, but also the provision of information and education on sexual and reproductive health [14, 15]. Additionally, it highlights the need to integrate reproductive health into national strategies and programs [14, 15]. Achieving this target requires addressing the prevention of teenage pregnancies and reducing the risks associated with early and unwanted pregnancies, which can lead to adverse obstetric outcomes. It is also essential to focus on preventing sexually transmitted diseases, including HIV and hepatitis, which disproportionately affect the health of adolescents, especially young girls. These efforts are essential for improving maternal and child health, as well as sexual and reproductive health indicators worldwide, particularly in regions heavily impacted by these issues. Therefore, it is imperative to identify the underlying causes that contribute to these risk factors among adolescents and develop effective interventions to mitigate them.

The sexual and reproductive health of Rwandan adolescents continue to be a major concern, with a range of challenges such as unplanned early pregnancies, unsafe abortions, risk of STIs and HIV/AIDS among young people who debuted sexual intercourses earlier [16–18]. The latest Rwanda Demographic Health Survey (RDHS) reveals that 5% of girls aged 15–19 have already started childbearing, with 4% having given birth and 1% being pregnant with their first child [19]. The occurrence of adolescent pregnancy stood at 6.11% (n = 553/9050) [20]. Over time, the rate of teenage pregnancy was recorded at 6.22% in 2010, 7.30% in 2015, and 5.41% in 2015 in terms of prevalence [20]. The research also suggested that residing in the Eastern region of Rwanda was associated with an increased likelihood of teenage pregnancy, with an adjusted odds ratio of 1.55 and a 95% confidence interval ranging from 1.078 to 2.24 [20]. These figures are alarming and indicate the urgent need for effective interventions to address the issue.

The number of underage girls getting pregnant in Rwanda has been on the rise in recent years, with official data showing that 19,832 girls got pregnant in 2018, up from 17,337 in 2017. The COVID-19 pandemic has further exacerbated the situation, with the number of teenage pregnancies increasing by 17.5% from 2017 to 2022 [21]. In the context of Rwanda, quantitative studies have reported that that teenage pregnancy is influenced by various factors

such as poverty, low educational attainment, limited exposure to sources of information about sexual and reproductive health, and poverty [20, 22]. Further research should be undertaken to investigate the additional risk factors to teenage pregnancy from adolescents' perspectives. Therefore, the current study aimed to provide a deeper understanding of how these factors impact the sexual and reproductive health of adolescents. By identifying these risk factors, effective interventions can be developed to prevent unplanned pregnancies and promote the overall sexual and reproductive well-being of adolescents. Furthermore, there is a scarcity of studies that explore the perspectives of adolescents regarding the factors affecting their navigation of the bodily and mental changes resulting from adolescence. Additionally, the study aimed to understand the support mechanisms available to them and the role of parents in reducing the risk factors of teen pregnancy and sexually transmitted infections including HIV/AIDS; particularly among male and female teenagers attending day-school secondary schools in the Eastern Province, Rwanda.

## Methodology

The present study was exploratory nature and employed a descriptive qualitative design. The reported findings are part of a larger qualitative research project undertaken to explore the perspectives of Rwandan secondary schools' students regarding the physiological and psychological changes occurring during adolescence and how these changes affect young adolescents' sexuality and reproductive health. The initial phase of the project centered on gathering young adolescents' understanding of the physiological and emotional changes taking place during different stages of adolescence. Additionally, it explored adolescents' personal experiences with these changes and examined the strategies they employed to navigate through them. The second phase which is reported in the current pater focused on factors affecting young adolescents' prevention of unwanted pregnancies and sexually transmitted infections including HIV/AIDS and what they need to maintain a healthy sexual and reproductive health. The utilization of a qualitative descriptive approach was favoured to attain a comprehensive depiction of adolescents' perspectives on the factors affecting their sexuality and sexual and reproductive health [23]. The research was carried out in six secondary schools, both in rural and urban areas, that were selected from three districts; Rwamagana, Kayonza, and Gatsibo districts located in the Eastern Province of Rwanda.

### Study population

The participants in this research consisted of both male and female adolescents who were pursuing their studies in twelve years of basic education in selected districts. This particular group of adolescents was considered because they can potentially face the vulnerability of participating in unsafe sexual activities, either as a means of personal exploration or due to the influence of older individuals. This susceptibility arises from the fact that they commute to and from school independently, without the presence of a caregiver. Furthermore, their age and lack of protection make them susceptible to being deceived or becoming victims of sexual assaults and advances from adults they come across during their daily travels.

The inclusion criteria for this study were set as follows: the participants had to be between the ages of 10 and 19 and residing in either rural or urban areas of Rwamagana, Kayonza, and Gatsibo districts. In order to participate in the study, each adolescent boy or girl was required to obtain an assent form, while their parents or guardians were required to provide consent. The study focused on six target twelve-year-basic education secondary schools, with two schools selected from both rural and urban areas in each district, allowing for a representativity

of the sample in order to balance the insights. The recruitment of participants and the data collection exercise for this study took place between 10th August 2022 and 8th December 2022.

## Sampling approach

Purposive sampling was used to identify and selected adolescents with predetermined inclusion criteria of this study [24]. The application of purposive sampling enables researchers in choosing participants with a greater understanding of the specific phenomenon under investigation [24]. Within the context of this study, purposive sampling was employed to identify and select adolescent boys and girls from each year of study and who were willing to provide their views and their personal experiences regarding the issues affecting their sexual and reproductive health within their respective residence areas.

## Sample size determination

The study targeted to involve 120 adolescent students including 60 girls and 60 boys to participate in the 12 focus group discussions each having 10 participants. The determination of sample size for qualitative studies involving focus group discussions still lacks an evidence-based guideline. Nevertheless, recent research indicates that two key factors, saturation and the identification of codes, play a crucial role in this process [25, 26]. Drawing from existing literature and considering practical constraints such as financial resources, it is recommended that conducting six focus groups with 60 female students and an additional six focus groups with 60 male students would likely capture over 95% of the necessary themes derived from the insights of participating adolescents [26, 27].

## Data collection

In order to gather comprehensive data from participating adolescent boys and girls, a semi structured interview guide with open questions asking about barriers young adolescents encounter as they navigate adolescence and what they need to leverage this transitional journey was employed to facilitate the focus group discussions. The interview guide was initially prepared in English and later translated into Kinyarwanda, the national language of Rwanda, to ensure inclusivity. By providing both English and Kinyarwanda versions of the interview guide, eligible participants were given the freedom to select the language they felt most comfortable with during the interview, accommodating their preferences.

To capture the viewpoints of adolescent boys and girls, a focus group discussion was utilized during the data collection process. By employing focused group interviews, participants were given the opportunity to express their viewpoints, fostering a safe and supportive environment among peers. To ensure a balanced and inclusive dynamic within the focused groups, the researchers organized them into clusters based on gender. This strategy aimed to mitigate power imbalances and promote a sense of comfort among the participants. By creating an environment where either adolescent boys or adolescent girls felt at ease, they were more likely to provide insightful responses regarding their perceptions and experiences with sexuality and sexual reproductive health. This approach further enhanced the participants' engagement and facilitated a deeper understanding of their unique perspectives within their community context.

The duration of the interview was estimated to be around 45–60 minutes. To prevent any potential distractions that could arise from note-taking during the interview, all discussions were recorded with the participants' consent. Furthermore, field notes and memos were taken to document the researcher's observations, and emotions regarding the environmental context and non-verbal expressions of the adolescents that may not have been adequately captured in

the recorded audios [28]. Those field notes guided the research team members to remember those impressions during data analysis and interpretations.

## Data management and analysis

The collected data were transcribed verbatim by the research team member. Since the research team members were proficiency in both used languages, this allowed them to translate all collected data from Kinyarwanda to English language. Dedoose software was used to manage and organize transcribed data to enhance the process of data analysis.

Inductive thematic analysis by Braun and Clarke [29] was followed to analyze and generate themes from collected transcripts. The authors implemented a six-step process of inductive thematic analysis, which involved familiarization, coding, generation of themes, reviewing of themes, defining and naming themes, and writing up. By utilizing thematic analysis methods, the research team was able to gain insights into the perspectives of adolescent boys and girls regarding the challenges affecting their sexual and reproductive health during adolescence.

## Measures undertaken to ensure quality of data

In order to establish the rigor of this study, four specific criteria were carefully considered: credibility, dependability, transferability, and confirmability. To ensure credibility, the researchers remained committed to accurately representing the perceptions of adolescents regarding their sexual and reproductive health changes. This was achieved by purposefully selecting participants who met specific criteria and were able to provide valuable insights into their physiological changes. The researchers engaged in extensive discussions to reach a consensus on the interpretation of the findings derived from the participants' interviews. This collaborative approach ensured that the outcomes were reliable and trustworthy.

To address the issue of dependability, the researchers developed an interview guide that was consistently followed in a sequential manner across all participants. This systematic approach allowed the research team to assess whether conducting the same study in the same context and with the same participants would yield consistent findings. By adhering to a standardized procedure, the researchers were able to establish the dependability of the study's results. In order to enhance the transferability of the study, a comprehensive description of the research methods was provided. This detailed account enables future researchers to replicate the study in different settings and environments. By sharing the methodology, other researchers can employ the same methods and compare their findings, thereby contributing to the overall understanding of the phenomena being studied.

**Ethical consideration.** The study underwent a rigorous approval process by the Institutional Review Board (IRB) at the University of Rwanda, College of Medicine and Health Sciences (Approval Notice: 303/CMHS IRB/2022). Subsequently, the ethical research committee of Eastern Province reviewed and granted ethical clearance for the study. To ensure compliance with ethical standards, a formal letter was composed and submitted to the District Executive, seeking permission for the researchers to interact with the research participants. In adherence to ethical guidelines, written informed consent and assent were obtained from the parents or guardians of the research participants prior to their involvement in the study. To maintain the confidentiality and anonymity of the participants, unique codes were assigned to each individual. Additionally, the participants were afforded the right to decline participation or withdraw from the study at any given time, emphasizing the importance of their autonomy and voluntary involvement.

## Results

A total of twelve focus group discussions were conducted as part of the study. These discussions were divided into two categories: six involving adolescent male students and six involving adolescent female students. The study included a total of 118 participants, comprising 58 boys and 60 girls. Additional details regarding the sample characteristics can be found in Table 1 below.

Table 1 indicates that the average age of the individuals involved was 16.6. A significant portion of the participants (91 out of 118) were pursuing ordinary level studies. More than half of the participants (57.6%) resided with their caregivers in the village after school, while 61% (n = 72) spent their holidays in the villages. Furthermore, 51 participants were not residing with both parents during the data collection period.

### Theme 1: Young adolescents' awareness of strategies to prevent STIs and unwanted pregnancies

This overarching theme encompasses two distinct sub-themes: the recognition among young adolescents of the importance of condoms in protecting against STIs and unintended

**Table 1. Sample characteristics.**

| Variable | | N | % |
|---|---|---|---|
| **Age in years** | 13.00 | 5 | 4.2 |
| | 14.00 | 19 | 16.1 |
| | 15.00 | 14 | 11.9 |
| | 16.00 | 30 | 25.4 |
| | 17.00 | 26 | 22.0 |
| | 18.00 | 24 | 20.3 |
| | Total | 118 | 100.0 |
| **Participants' year of study** | 1.00 | 39 | 33.1 |
| | 2.00 | 33 | 28.0 |
| | 3.00 | 19 | 16.1 |
| | 4.00 | 9 | 7.6 |
| | 5.00 | 13 | 11.0 |
| | 6.00 | 5 | 4.2 |
| | Total | 118 | 100.0 |
| **Where she/he lives after school** | Village | 68 | 57.6 |
| | Town | 50 | 42.4 |
| | Total | 118 | 100.0 |
| **Where she/he lives during holidays** | Village | 72 | 61.0 |
| | Town | 46 | 39.0 |
| | Total | 118 | 100.0 |
| **The persons who care for the young child** | Both parents | 67 | 56.8 |
| | Only lives with the mother | 46 | 39.0 |
| | Only lives with the father | 2 | 1.7 |
| | Lives with relatives | 3 | 2.5 |
| | Total | 118 | 100.0 |
| **Religion** | Christians | 102 | 86.4 |
| | Islam | 16 | 13.6 |
| | Total | 118 | 100 |

pregnancies, as well as young adolescents' understanding of the adverse consequences associated with engaging in risky sexual behaviors.

*Subtheme 1*: *A condom can prevent unplanned pregnancies and sexually transmitted infections, but we lack parental and practical guide on how to use it*. The majority of participants reported that parental guidance and consistent reminders are necessary for both boys and girls on the significance of using condoms during sexual intercourse. They emphasized that some of the teenage girls often become pregnant due to limited knowledge about condom use.

"*You know for most of us boys, we don't even mind much about using condoms. But if parents should provide education to girls about condom use and even buy some for their daughters just in case, I think the cases of unintended pregnancies we see here would decrease.*" (Male participant 3, Kayonza FGD)

*Subtheme 2*: *Young adolescents are aware of the negative outcomes poorly managed sexuality during adolescence*. All participating adolescents were aware of the consequences they can face if they mismanage their adolescence. Adolescent males possess knowledge regarding the potential consequences of engaging in unprotected sexual intercourse, such as the possibility of impregnating their female counterparts. Additionally, they are cognizant of the various risks associated with unprotected sex, including the transmission of sexually transmitted infections, notably HIV/AIDS. Moreover, they are well-informed about the emotional and psychological burdens that teenage pregnancy imposes on both young males and females, as well as the social stigma attached to HIV infection, which can contribute to feelings of depression and contemplation of self-harm.

"*Unprotected sex carries a risk for boys as well because we too can get HIV. Another consequence is when you have sex with a woman older than you, she may transmit sexually transmitted infections to you. When you learn that you have those infections or HIV, you can become depressed to the point that you indulge in abusing drug substances thinking that we are getting relief. However, the end result would be harming our health and also leading to suicidal ideas. Another issue is to manage the pressure of assuming family and parental responsibilities at young age which affect your future health.*" (Male participant 5, Mukarange FGD).

The anguish and dilemma faced by young boys intensify when they contract HIV from a girlfriend or a trusted female figure, such as a close relative.

"*You feel heartbroken, and it is not even easy to report such cases to the police because of the misconception that reporting such cases would be an embarrassment for the extended family. So, because of sorrow and flashbacks on how you have fallen in that mistake of sleeping with her and get HIV, you can end up infecting others intentionally as a way of revenge.*" (Male participant 8, Kayonza FGD).

The issue of assuming pregnancy and parental responsibilities at a young age was a topic of concern among most adolescent boys. During discussions, participants highlighted various challenges associated with this situation, including the lack of preparedness to become parents while still relying on their own parents for support. Additionally, it was noted that impregnating girls while attending school could result in both the young boy and girl dropping out of their education, which was an undesirable consequence. The viewpoints of young girls aligned with those of their male peers. They believed that if boys impregnate young girls, they may be compelled to enter into early parenthood and take on the responsibilities of being a parent.

"*It happens that a young boy does unprotected sex out of exploration and curiosity and then impregnate the young girl. . . When it happens, the young boys' parents instead of supporting him, tell him to assume the parenting and family care responsibilities while still very young. That's when you see the young boy dropping out school to assume family responsibilities.*" (Male participant 3, Kiziguro FGD)

Adolescent girls who fail to navigate their teenage years effectively may encounter a range of adverse outcomes, such as unintended pregnancies, heightened vulnerability to sexually transmitted infections (including HIV), substance misuse, and engagement in sex work. In light of these circumstances, young boys reiterated that teenage pregnancy impacts young girls in various ways, such as familial rejection, the onset of depression leading to contemplation of suicidal thoughts, and resorting to secretive abortions, as evidenced by the following quotations.

"*Most of the time when girls become pregnant, they keep it secret because some parents do not want to hear their daughters being pregnant. Others may find it shameful to bear and raise that baby in the future. As a result of hiding the pregnancy, some young girls may decide to have clandestine abortion of course which is threatening because she may die. If she does not die, still her reproductive system mainly her womb can be destroyed.*" (Female participant 8, Kayonza FGD).

"*There are some girls who become pregnant and keep it a secret. They hide that from anyone, even from their parents. During their pregnancy, they do not take care of themselves, and they may not eat properly. On the day of delivery, such girls don't go to the health facilities and may risk dying giving birth or their babies may die in the process.*" (Female participant 5, Kiziguro FGD).

Female participants reported that the occurrence of unplanned pregnancies and the subsequent actions they take to avoid the responsibilities of parenting without any form of support. These actions often involve attempting secretive abortions.

"*You know not all our families are rich, so when you get pregnant and you don't get support from your parents and the baby's father support, surely you suffer quite a lot when upbringing the baby as a single mother.*" (Female participant 1, Mulinga FGD).

The impact of unplanned pregnancy on the lives of female participants was vividly expressed, especially during the process of childbirth.

"*Teen pregnancy for some young girls can result into caesarean during the time of delivery which is a risk itself. A girl may also have prolapsed uterus during delivery, and this may result into infertility later.*" (Female participant 7, Mulinga FGD).

Female participants also reported that young girls who do not receive sufficient support may also experience depression, discontinue their education, and face the harsh reality of extreme poverty.

"*My elder sister has so far got two kids and she has dropped out of school. Now, she is depressed and whenever we try to find her a job, she doesn't want to work.*" (Female participant 7, Mukarange FGD).

"*One of my classmates got pregnant and she suffered a lot. Her parents were very poor and were even finding it difficult to find money for renting. After giving birth, she was obliged to leave her parents and lead a life of her own. It was heartbreaking seeing her because at times she might not obtain food and you know when you have a baby, you need sufficient food so that you can breastfeed the baby. She was starving and it was also difficult for her to get clothes for herself and the baby.*" (Female participant 5, Kiziguro FGD).

### Theme 2: Barriers affecting young people's sexual and reproductive health during adolescence

The second theme encompasses three separate sub-themes, namely: a) Insufficient engagement of parents in educating their young adolescents about sexual and reproductive health, b) misconceptions surrounding sexuality during adolescence, and c) inadequate knowledge regarding the menstrual cycle.

*Subtheme 1*: *Limited parental involvement in their young adolescents' sexual and reproductive health education*. The lack of parental guidance regarding sexual and reproductive matters has been identified by the majority of participants as a significant challenge for young boys and girls in the Eastern province of Rwanda. Participants reported that this absence of parental voice contributes to the various risks that adolescents in the region face. The main barriers to parental involvement reported by participants include conflicts within families and a lack of understanding about sexuality issues. The participants also emphasized that conflicts within families have a detrimental impact on their sexual and reproductive health.

"*When there is conflict within a family, the parents do not educate their children about sexual and reproductive health. Since a young child has no one to provide her with information or even warn her about certain bad behaviours, he/she takes decisions themselves. For us girls, that's when we will end up sleeping with boys and get unintended pregnancies.*" (Female participant 9, Kiziguro FGD).

Female participants reported that when parents are absent, especially in the case of young girls, they may leave their families in search of informal employment opportunities in their local communities or urban areas. Unfortunately, this search for employment can expose these young girls to the risk of sexual exploitation. Additionally, they noted that parents often neglect the informational needs of young boys regarding sexual and reproductive health. They mentioned that parents still consider these topics as taboo and lack the necessary knowledge to provide guidance to their children.

"*The challenge is that some young people see the pubertal changes and we fear to inform our parents. When we tell them about the changes we saw, they end up retorting that it is not time for us to know such obscene things because they are still children. Oftentimes, when parents answer you like that you keep quiet and anticipate to get information elsewhere*". (Female participant 3, Kayonza FGD).

The participants also highlighted the lack of attention from parents towards their sexual and reproductive health, especially boys, as a contributing factor to the rise in unintended teenage pregnancies within their community's health as highlighted in quotes below.

"*There are some parents who even fear their teen children because perhaps they do not know where to start the conversation around for example sexuality concerns.*" (Female participant 6, Mulinga FGD).

Another female participant corroborated the fact that parents shy away from discussing topics related to sexuality and reproductive health with their children.

"*When you have hostile mother, she doesn't give you time to discuss about reproductive health issues. You know some of our parents do not even ask us if we have seen our periods or even about life in general because they take that as normal things. Whether we do sex or do not, they don't care.*" (Female participant 6, Kayonza FGD).

*Subtheme 2*: *Misconception surrounding sexuality during adolescence.* Although a significant number of participants reported strategies to navigate the various changes taking place during adolescence, some of them noted that some girls may misinterpret the presence of acne for example as a reason to engage in sexual activities. Male participants reported that young male adolescents often may receive inaccurate information about circumcision from their peers or the general public, which could potentially lead them to engage in unprotected sex.

"*Outside here there is a misconception that when recovering from the circumcision wound, you should seek a girl to sleep with in order to reduce the sores where the out skin of the male sex has been removed.*" (Male participant 7, Mukarange FGD).

Other male participants expressed that some of their peers give them wrong information about sexuality during adolescence.

"*You know, your friends can tell you if you do sex, you will feel relieved from stress. They tend to only focus on the good things you will benefit from and do not tell you about the negative side. Most of the time, if you don't accept what they are doing, they call you naïve or a coward.*" *(Male* participant 8, Munyiginya FGD).

Social media, as an alternative information source, has the potential to mislead young individuals. It is concerning that some young people may exploit these platforms to search for explicit content, including sexuality-related information and pornography.

"*One day I gave my smartphone to my cousin. She opened YouTube to search for something using my internet bundle. You know what, at the end of the day, when I checked the search history, I was surprised to see that she was search videos around sexuality from Rwanda.*" (Female participant 4, Kayonza FGD).

*Subtheme 3*: *Lack of adequate information on menstrual cycle.* In response to inquiries regarding the possibility of a girl becoming pregnant, a majority of young boys and girls expressed a lack of adequate knowledge on the matter. Upon further probing, a small number of participants provided varied responses, indicating the necessity for enhanced education on the human reproductive cycle. Some female participants who attempted to answer unanimously suggested that a girl can conceive within a range of one to eight days following her menstrual bleeding and four days preceding her periods. Male participants on the other hand stated that a boy could potentially impregnate a girl at any time, regardless of her monthly cycle.

"*When we were in class if I remember well, they taught us that a girl may not be pregnant within the four days after having periods, but she can get pregnant four days before she sees her periods. They also told us that you can even get pregnant during periods if they do sex.*" (Female participant 6, Mukarange FGD).

### Theme 3: Informational needs for young people during adolescence

Participants highlighted a number of informational needs to leverage their sexual and reproductive health. Participants underlined that young girls are more vulnerable during adolescence due to the potential consequences of early unintended pregnancies, which can hinder their future prospects. In order to address this vulnerability, participants underscored the importance of providing comprehensive education on sexuality and reproductive health, including information on preventing unplanned pregnancies and sexually transmitted infections.

"*For me, I think that a young girl needs to know that anytime she does sex, she may become pregnant which in turn can affect her future. Therefore, young girls need to know that, and one way is to empower them with information that can enable them to resist risky sexual behaviours.*" (Female participant 2, Kiziguro FGD).

Additionally, they suggested that young girls should also be educated on the proper use of condoms, benefits of using them as a means of protection when abstinence is not an option and saying no to any romantic relationships luring them to unprotected sexual intercourses. The female participants corroborated their male counterparts' viewpoints. Female participants stressed the importance of parents providing constant information and material support to protect them from sexual exploitation.

"*. . .. a young girl should be advised by her parents on being cautious. I say this because during adolescence, we are vulnerable because we can be misled by men who might need to exploit us sexually. So, when we don't have sufficient support, we can end up being lured by their fake gifts and then get pregnant or even get sexually transmitted infections.*" (Female participant 4, Munyiginya FGD).

Furthermore, several female participants emphasized the importance of providing young girls with education on effective reporting of incidents of sexual violence committed against them.

"*During adolescence, some girls may be lured into trusting men particularly those who give them lift or other material things. Such men may one day demand to be paid back through sleeping with them. If a young girl refuses, such men can rape her. In case this incident happens, a young girl needs to be aware of the reporting processes and be empowered to inform her parents for timely interventions*." (Female participant 8, Kiziguro FGD).

All participants have expressed that one of the primary informational requirements to prevent teen pregnancy is obtaining knowledge about the menstrual cycle.

"*Honestly, even though we studied reproductive system in class, we do not really have information about a woman's monthly cycle. Particularly, we don't know which period of the cycle a girl can be pregnant if you sleep with her.*" (Male participant 9, Kayonza FDG).

Female participants further emphasized the need to equally educate young boys about sexual and reproductive health matters, highlighting that these topics should not be exclusive to girls.

"*You know, when they mention reproductive health, some of the young boys will interpret it as a girl's thing, you know conception and menstruation things like that. They often don't feel concerned. Therefore, to limit such attitudes boys too need to be educated about sexual and reproductive health. This can be a protective thing for us girls because informed young boys can protect us.*" (Female participant 1, Munyiginya FGD).

## Discussion

The objective of our study was to identify the factors that facilitate and/or hinder the SRH of rural teenage secondary school students in the Eastern Province of Rwanda. Additionally, our study aimed to explore the specific informational requirements related to SRH of this population, as well as to investigate the existing support systems that aid in promoting healthy SRH practices during adolescence. Furthermore, we sought to examine the contributions of parents and caregivers in mitigating the risks associated with unwanted teen pregnancies and STIs among young adolescents in rural secondary schools in the Eastern Province of Rwanda.

Our study indicate that adolescents, both male and female, attending secondary schools in the eastern province of Rwanda are aware of the significance of condoms in preventing unintended teenage pregnancies. However, they face a lack of constant parental guidance and support when it comes to the practical application of condoms and the necessary sensitization regarding their usage. Although our research did not include input from parents regarding the sexual and reproductive health of young adolescents, several studies have documented that Rwandan culture considers discussions on sexual health particularly condom use to be a sensitive subject when it comes to children on grounds that it may encourage them to be promiscuous [30–32]. This finding aligns with similar studies conducted in South Africa, Tanzania, and Nigeria where parents expressed discomfort and deemed it inappropriate to engage in conversations about puberty, condom use, sexually transmitted infections, and contraceptives due to the belief that their children are not yet mature enough to handle such topics [33–35].

Our study suggests that adolescent boys and girls are aware of the potential repercussions linked to engaging in unprotected sexual activities during their teenage years, such as sexually transmitted infections, HIV, and the challenges of early parenthood despite the fact they at times get overwhelmed by sexual desire. Moreover, the study found that teenagers attending secondary schools in the eastern region of Rwanda are cognizant of and apprehensive about the adverse emotional and psychological effects associated with the negative outcomes of unprotect sex. Our research validates the results of a similar study carried out in Rwanda, indicating that despite experiencing a strong physical desire for sexual activity, young individuals are aware of the serious consequences that may arise from engaging in unprotected sexual intercourse [36]. Therefore, we suggest that interventions targeting the prevention of STIs, HIV transmission, and unintended pregnancies should be based on the current awareness that young individuals possess regarding the negative outcomes associated with risky sexual behaviors. Moreover, these interventions should empower them by highlighting the advantages of practicing abstinence or utilizing condoms whenever they engage in sexual activities.

We found that adolescents in secondary schools located in the Eastern Province of Rwanda are aware of the challenges faced by young girls due to unintended pregnancies. These difficulties include potential rejection from their guardians, feelings of depression, discontinuation of

education, resorting to unsafe abortion practices, experiencing complicated childbirth, and facing financial hardship post-delivery for those who decide to keep the baby. Despite being aware of the consequences of unprotected sex on girls, young adolescents continue to participate in such risky behaviors.

Our study revealed that inadequate parental engagement in the daily lives of their adolescent children, attributed to being unapproachable when it comes to communicating about sexual and reproductive health, work commitments, family disputes, and the perception of sexuality and reproductive health changes during puberty as taboo subjects, has a detrimental impact on young people in the Eastern province of Rwanda. The limited parental communication with young adolescents about sexuality and reproductive health may expose them to unsafe sexual behaviors that can result in sexually transmitted infections (STIs), HIV, and unintended pregnancies. Our research enhances the results of numerous studies on the obstacles impacting the sexual and reproductive health of adolescents in Rwanda and other regions of Eastern and Southern Africa [33, 34, 37–41]. These studies have revealed that parents tend to be uncooperative when it comes to addressing sexual and reproductive health matters with their children [30, 33, 37, 42, 43]. This behavior is often driven by the perception that young boys and girls may be too immature to engage in romantic relationships, leading parents to believe that discussions on sexual and reproductive health topics are unnecessary [30, 34, 40, 41].

## Strengths and limitations

One of the main strengths of our study lies in the fact that the sample consisted of young adolescent boys and girls who were in a favorable position to provide comprehensive information regarding the phenomenon under investigation. However, it is important to acknowledge that this study does have certain limitations. Firstly, our sample only included young adolescents who were attending school, specifically from six secondary schools. As a result, the transferability of the findings may be restricted to this particular population, and therefore, the perspectives expressed by the participants may not be representative of all young individuals in the Eastern Province of Rwanda, including those who are not enrolled in school. This omission of young people who were not attending basic secondary education schools means that their experiences were not accounted for in our study. Additionally, we did not employ triangulation in our data collection methods, such as conducting individual interviews. Consequently, it is possible that some participants may have chosen to conceal their personal experiences with sexuality and the barriers they faced in terms of sexual and reproductive health during adolescence. This limitation resulted in some young individuals discussing general barriers that young people encounter, without providing specific context based on their own experiences.

The primary emphasis of the current study was on unintended pregnancy and its consequences, as well as sexually transmitted infections, including HIV/AIDS. Other essential components of sexual and reproductive health, such as life skills, sexual and reproductive health rights, responsibility, and the promotion of assertiveness among young adolescents, were not addressed. Consequently, further research should explore these aspects of sexual and reproductive health in order to develop a holistic sexual and reproductive education program that will empower young individuals in the Eastern Province of Rwanda to avoid unwanted pregnancies and STIs, including HIV/AIDS.

## Conclusion

This research has recognized obstacles that influence the sexual and reproductive well-being of young people, particularly the insufficient participation of certain parents in issues concerning

sexuality and reproductive health changes experienced during adolescence. The absence of parental involvement impedes adolescents in managing the emotional, psychological, and physical transformations of adolescence, hindering their ability to make informed choices regarding their health and providing them with the necessary support to avoid negative outcomes like unplanned pregnancies, substance abuse, sexually transmitted infections, and HIV/AIDS.

In order to improve the sexual and reproductive well-being of young people, the Ministry of Health needs to work with the Ministry of Gender and Family Promotion and other relevant stakeholders to develop educational resources centered on sexual and reproductive health. These materials should be aimed at educating young adolescents in secondary schools about the physical, physiological, and emotional changes that take place during adolescence, which may potentially lead to engaging in unsafe sexual practices and subsequently result in unintended teenage pregnancies.

Due to the qualitative nature of this study, it was not possible to gather quantitative data that would allow us to accurately assess the magnitude of the issue regarding the lack of parental involvement in educating adolescent boys and girls about sexual and reproductive health. Consequently, additional research is required in order to address this gap.

## Supporting information

**S1 File.**
(DOCX)

## Acknowledgments

We would like to acknowledge the financial support from Center for International Reproductive Health Training at the University of Michigan, USA (CIRHT-UM) who provided a seed grant to facilitate data collection and analysis that contributed to this manuscript.

## Author Contributions

**Conceptualization:** Thierry Claudien Uhawenimana, Kellen Muganwa, Marie Chantal Uwimana, Marie Grace Sandra Musabwasoni, Olive Tengera, Joy Bahumura.

**Data curation:** Thierry Claudien Uhawenimana, Kellen Muganwa, Francoise Mujawamariya, Richard Nsengiyumva.

**Formal analysis:** Thierry Claudien Uhawenimana, Richard Nsengiyumva.

**Funding acquisition:** Thierry Claudien Uhawenimana, Kellen Muganwa.

**Investigation:** Kellen Muganwa, Marie Grace Sandra Musabwasoni, Emmerance Uwingabire, Francoise Mujawamariya, Richard Nsengiyumva, Oliva Bazirete.

**Methodology:** Thierry Claudien Uhawenimana, Kellen Muganwa, Marie Chantal Uwimana, Marie Grace Sandra Musabwasoni, Olive Tengera, Joy Bahumura, Joella Mukashyaka, Jackline Mukakamanzi, Emmanuel Habyarimana, Innocent Ngerageze, Hellen Nwanko Chinwe, Emmerance Uwingabire, Francoise Mujawamariya, Richard Nsengiyumva, Oliva Bazirete.

**Project administration:** Kellen Muganwa.

**Software:** Thierry Claudien Uhawenimana.

**Supervision:** Thierry Claudien Uhawenimana, Kellen Muganwa, Richard Nsengiyumva.

**Validation:** Thierry Claudien Uhawenimana, Kellen Muganwa, Richard Nsengiyumva, Oliva Bazirete.

**Visualization:** Thierry Claudien Uhawenimana.

**Writing – original draft:** Thierry Claudien Uhawenimana, Kellen Muganwa, Marie Chantal Uwimana, Marie Grace Sandra Musabwasoni, Olive Tengera, Joy Bahumura, Joella Mukashyaka, Jackline Mukakamanzi, Emmanuel Habyarimana, Innocent Ngerageze, Hellen Nwanko Chinwe, Emmerance Uwingabire, Francoise Mujawamariya, Richard Nsengiyumva, Oliva Bazirete.

**Writing – review & editing:** Thierry Claudien Uhawenimana, Kellen Muganwa, Marie Chantal Uwimana, Marie Grace Sandra Musabwasoni, Olive Tengera, Joy Bahumura, Joella Mukashyaka, Jackline Mukakamanzi, Emmanuel Habyarimana, Innocent Ngerageze, Hellen Nwanko Chinwe, Emmerance Uwingabire, Francoise Mujawamariya, Richard Nsengiyumva, Oliva Bazirete.

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
