## [Decision Letter · Decision Letter 0]

10 May 2024

PONE-D-24-09798“We need parental guidance to lead safe and healthy sexual and reproductive health during adolescence”: Perspectives of young adolescents from the Eastern Province of RwandaPLOS ONE

Dear Dr.  UHAWENIMANA,

Thank you for submitting your manuscript to PLOS ONE. After careful consideration, we feel that it has merit but does not fully meet PLOS ONE’s publication criteria as it currently stands. Therefore, we invite you to submit a revised version of the manuscript that addresses the points raised during the review process. A point by p[oints responses are required for all comments raised. Please submit your revised manuscript by Jun 24 2024 11:59PM. If you will need more time than this to complete your revisions, please reply to this message or contact the journal office at plosone@plos.org. Please include the following items when submitting your revised manuscript:A rebuttal letter that responds to each point raised by the academic editor and reviewer(s). You should upload this letter as a separate file labeled 'Response to Reviewers'.A marked-up copy of your manuscript that highlights changes made to the original version. You should upload this as a separate file labeled 'Revised Manuscript with Track Changes'.An unmarked version of your revised paper without tracked changes. You should upload this as a separate file labeled 'Manuscript'.

We look forward to receiving your revised manuscript.

Kind regards,

Yitagesu Habtu Aweke, Ph.D

Academic Editor

PLOS ONE

Journal Requirements:

Reviewers' Comments:

**Comments to the Author**

1. Is the manuscript technically sound, and do the data support the conclusions?

Reviewer #1: Partly

Reviewer #2: Yes

2. Has the statistical analysis been performed appropriately and rigorously? 

Reviewer #1: N/A

Reviewer #2: N/A

3. Have the authors made all data underlying the findings in their manuscript fully available?

Reviewer #1: No

Reviewer #2: No

4. Is the manuscript presented in an intelligible fashion and written in standard English?

Reviewer #1: Yes

Reviewer #2: No

5. Review Comments to the Author

Reviewer #1: As stated, this article presents part of the results of a more comprehensive study. Maybe this issue has caused ambiguity and some questions:

- What exactly did the research team mean from the “perspectives of young adolescents”?

- How was the interview and managing the meetings guide compiled and what questions did it include?

- Certainly, significant data was obtained from the 12 focus group discussion sessions. What are the special features of the three presented “themes” that are reported separately from other research findings? Although the title of the third theme is also very general and non-specific.

- It seems that the researchers' main focus was on unwanted pregnancy (also its outcomes) and HIV/AIDS. Based on this, it was concluded that the awareness of adolescents was acceptable. But with the "comprehensive sexuality education" perspective, we know that having awareness alone is not enough to protect teenagers. Are concepts such as life skills, sexual and reproductive health rights, responsibility, assertiveness and saying no, etc. addressed in this study?

Reviewer #2: Thank you for this manuscript investigating themes associated with risky sexual behavior in Rwandan adolescents. There are a few things that would strengthen this manuscript, including being careful about terminology, specific about approach, and including some more context in the conclusion about what programs have (or have not been successful) and their target audience. There are also minor spelling and grammar issues throughout, the manuscript would benefit from a close review to correct these.

Abstract:

“and 4% are currently pregnant with their first child”

Intro:

- This conclusion in the first paragraph feels a bit abrupt, consider removing “thus” and making it the first sentence of the next paragraph “Thus, in order to promote the well-being of adolescents, it is essential to safeguard their sexual and reproductive health.”

- Rather than “Research have also revealed that” should be “Research has also revealed that”

- This reads as though >6 million pregnancies resulted in unsafe abortions … “These unintended pregnancies resulted in 12 million births, and 55% of them ended in unsafe abortions[9].”

- This sentence is confusing, were approx. half unintended or approx half from sub-saharan Africa? “In 2019 alone, it was estimated that 21 million pregnancies occurred in this age group, with approximately half of them being unintended were from Sub-Saharan Africa[10].”

- Consider defining unintended pregnancies in the intro – specifically these can be unwanted, or can be unplanned (which are qualitatively different). Not sure if that distinction will be made in the qualitative data but unplanned is not necessarily unwanted.

- Also may be useful to define sexual and reproductive health – think this is encompassing access to care, access to contraception, prevention of STDs? As I read through the intro, I am starting to think the specific research question is about risk factors for teenage pregnancy, not a the broader concept of sexual and reproductive health. Obviously its helpful to situation one within the other, but it would be helpful to re-frame this so its clear what the article addresses, and why that is important in the larger picture for adolescents.

Methods:

- For inclusion criteria “capable of expressing themselves” I am not sure how that was assessed or defined. Then it states “Within the context of this study, purposive sampling was employed to identify and select adolescent boys and girls who were capable of articulating their viewpoints and recounting their personal experiences regarding the issues affecting their sexual and reproductive health within their respective residence areas.” But this makes it sound like only articulate adolescents were included, which makes me concerned about bias …

- How was sample size determined?

Discussion/conclusion:

- Were gender differences seen in the types (or ways) themes were discussed?

- Nevertheless, our research did not address the determinants of risky sexual behavior among young people attending secondary schools in the Eastern province of Rwanda.

- “consisted of sexually active young individuals” I don’t think this was inclusion criteria?

- In the conclusion, I question whether a program for parents is the best approach given the findings. You also haven’t cited any programs that have done this successfully (or not), but given the cultural factors and the difficulty of accessing parents for a training, I would propose that providing information and training directly to students in schools may be more effective (and much easier logistically).

6. PLOS authors have the option to publish the peer review history of their article (what does this mean?). If published, this will include your full peer review and any attached files.

Reviewer #1: **Yes: **Seyed Ali Azin

Reviewer #2: No

---

## [Author Response · Author response to Decision Letter 0]

25 May 2024

Comments How the comments have been addressed

Academic Editor

We did not change our financial disclosure because the way it is stated reflected our situation.

Not applicable as our study was qualitative.

Thank you for raising this important issue. In the revised manuscript we followed the journal’s style requirements throughout. In addition, the caption of the supporting information files were renamed according to the Journal’s requirements.

Reviewer 1

As stated, this article presents part of the results of a more comprehensive study. Maybe this issue has caused ambiguity and some questions:

- What exactly did the research team mean from the “perspectives of young adolescents”?

- How was the interview and managing the meetings guide compiled and what questions did it include?

- Certainly, significant data was obtained from the 12 focus group discussion sessions. What are the special features of the three presented “themes” that are reported separately from other research findings? Although the title of the third theme is also very general and non-specific. Thank you for raising this important issue regarding the fact that this study presents part of the results of a larger study. We indicated in the methodology that:

‘The initial phase of the project centered on gathering young adolescents’ understanding of the physiological and emotional changes taking place during different stages of adolescence. Additionally, it explored adolescents’ personal experiences with these changes and examined the strategies they employed to navigate through them. The second phase which is reported in the current pater focused on factors affecting young adolescents’ prevention of unwanted pregnancies and sexually transmitted infections including HIV/AIDS and what they need to maintain a healthy sexual and reproductive health.’ (see page 7)

In order to match our study with the results and the discussions thereof, we changed the title for this manuscript to:

‘Factors affecting the prevention of unwanted pregnancies among young adolescents in secondary schools in the Eastern Province of Rwanda: An explorative qualitative study’

We have also restructured the reporting of the results. In the current revised manuscript version, theme 1 has been subdivided into two sub-themes, theme 2 into three sub-themes, and theme 3 with no sub-themes. This was done after adding more results to respond to the concerns raised by the reviewer. (For details, see results chapter from pages 13-21)

- It seems that the researchers' main focus was on unwanted pregnancy (also its outcomes) and HIV/AIDS. Based on this, it was concluded that the awareness of adolescents was acceptable. But with the "comprehensive sexuality education" perspective, we know that having awareness alone is not enough to protect teenagers. Are concepts such as life skills, sexual and reproductive health rights, responsibility, assertiveness and saying no, etc. addressed in this study? Thank you for this important commentary. In the limitations of this study, we acknowledged that our study did not cover concepts like life skills, sexual and reproductive health rights, and assertiveness and we recommended that further research is needed to investigate these concepts in the context adolescents’ sexual and reproductive health education. (see page 24 for details)

Reviewer 2

There are a few things that would strengthen this manuscript, including being careful about terminology, specific about approach, and including some more context in the conclusion about what programs have (or have not been successful) and their target audience. Thank you for the constructive comment. In the revised version, the raised issues were taken into consideration.

Abstract:

“and 4% are currently pregnant with their first child” Addressed. See page 2

- This conclusion in the first paragraph feels a bit abrupt, consider removing “thus” and making it the first sentence of the next paragraph “Thus, in order to promote the well-being of adolescents, it is essential to safeguard their sexual and reproductive health.” Addressed. See page 4

- Rather than “Research have also revealed that” should be “Research has also revealed that” Addressed. See page 5

- This reads as though >6 million pregnancies resulted in unsafe abortions … “These unintended pregnancies resulted in 12 million births, and 55% of them ended in unsafe abortions[9].” Addressed. See page 5

- This sentence is confusing, were approx. half unintended or approx half from sub-saharan Africa? “In 2019 alone, it was estimated that 21 million pregnancies occurred in this age group, with approximately half of them being unintended were from Sub-Saharan Africa[10].”

- Consider defining unintended pregnancies in the intro – specifically these can be unwanted, or can be unplanned (which are qualitatively different). Not sure if that distinction will be made in the qualitative data but unplanned is not necessarily unwanted. The ambiguation in this sentence has been addressed by removing the phrase ‘these unintended pregnancies resulted in 12 millions births’ (see page 5)

- Also may be useful to define sexual and reproductive health – think this is encompassing access to care, access to contraception, prevention of STDs? As I read through the intro, I am starting to think the specific research question is about risk factors for teenage pregnancy, not a the broader concept of sexual and reproductive health. Obviously its helpful to situation one within the other, but it would be helpful to re-frame this so its clear what the article addresses, and why that is important in the larger picture for adolescents. Thank you for underlining the need to define the terms ‘unintended pregnancies, and sexual and reproductive health’ in the context of this study. We provided that definition on page 4

- For inclusion criteria “capable of expressing themselves” I am not sure how that was assessed or defined. Then it states “Within the context of this study, purposive sampling was employed to identify and select adolescent boys and girls who were capable of articulating their viewpoints and recounting their personal experiences regarding the issues affecting their sexual and reproductive health within their respective residence areas.” But this makes it sound like only articulate adolescents were included, which makes me concerned about bias … Thank you for the advice. We have removed the phrase ‘capable of expressing themselves. See page 8 for details.

- How was sample size determined? Thank you for this observation. We have added a section about sample size determination. See page 9 for details.

Discussion/conclusion:

- Were gender differences seen in the types (or ways) themes were discussed? Thank you. Since our primary objective was to identify factors affecting the prevention of unintended pregnancies and STIs, we did not discuss the gender differences in the themes discussed because we thought that there no major differences from the viewpoints of male and female adolescents. 

- Nevertheless, our research did not address the determinants of risky sexual behavior among young people attending secondary schools in the Eastern province of Rwanda.

- “consisted of sexually active young individuals” I don’t think this was inclusion criteria? Thank you for this observation. The comment was considered and we removed the two sentences that focused on that aspect(see page 23).

- In the conclusion, I question whether a program for parents is the best approach given the findings. You also haven’t cited any programs that have done this successfully (or not), but given the cultural factors and the difficulty of accessing parents for a training, I would propose that providing information and training directly to students in schools may be more effective (and much easier logistically). Thank you for your observation. As you highlight it, educational interventions targeting students in schools would be more effective and much easier logistically. We corrected accordingly. (See page 25)

---

## [Decision Letter · Decision Letter 1]

2 Jul 2024

PONE-D-24-09798R1Factors affecting the prevention of unwanted pregnancies among young adolescents in secondary schools in the Eastern Province of Rwanda: An explorative qualitative studyPLOS ONE

Dear Dr. UHAWENIMANA,

Thank you for submitting your manuscript to PLOS ONE. After careful consideration, we feel that it has merit but does not fully meet PLOS ONE’s publication criteria as it currently stands. Therefore, we invite you to submit a revised version of the manuscript that addresses the points raised during the review process.

Please include the following items when submitting your revised manuscript:A rebuttal letter that responds to each point raised by the academic editor and reviewer(s). You should upload this letter as a separate file labeled 'Response to Reviewers'.A marked-up copy of your manuscript that highlights changes made to the original version. You should upload this as a separate file labeled 'Revised Manuscript with Track Changes'.An unmarked version of your revised paper without tracked changes. You should upload this as a separate file labeled 'Manuscript'.We look forward to receiving your revised manuscript.

Kind regards,

Yitagesu Habtu Aweke, Ph.D

Academic Editor

PLOS ONE

Reviewers' comments:

Reviewer's Responses to Questions

**Comments to the Author**

1. If the authors have adequately addressed your comments raised in a previous round of review and you feel that this manuscript is now acceptable for publication, you may indicate that here to bypass the “Comments to the Author” section, enter your conflict of interest statement in the “Confidential to Editor” section, and submit your "Accept" recommendation.

Reviewer #1: All comments have been addressed

Reviewer #2: (No Response)

2. Is the manuscript technically sound, and do the data support the conclusions?

Reviewer #1: Yes

Reviewer #2: Yes

3. Has the statistical analysis been performed appropriately and rigorously? 

Reviewer #1: N/A

Reviewer #2: Yes

4. Have the authors made all data underlying the findings in their manuscript fully available?

Reviewer #1: Yes

Reviewer #2: No

5. Is the manuscript presented in an intelligible fashion and written in standard English?

Reviewer #1: Yes

Reviewer #2: Yes

6. Review Comments to the Author

Reviewer #1: The revised article provides more information and is more coherent. Mentioning details about the questions and topics of focused group discussions could help improve the article.

Reviewer #2: Thank you for this revised manuscript. Several revisions I think were helpful for clarity. I do have a few additional comments, below.

• This edit: “an unwanted pregnancy is described as a pregnancy that occurs before they have reached physical and mental maturity to fully understand the implications of their actions and provide informed consent.” With current edits, I think this sentence can just be dropped.

• Paragraph 2 of the intro, there are several acronyms used that are not spelled out.

• This section in the intro is still a bit confusing: “In 2019 alone, it was estimated that 21 million pregnancies occurred in this age group, with approximately half of them being unintended were from Sub-Saharan Africa[14].” Were approximately half unintended or half were from SSA?

• Similarly, I am not sure what “These unintended pregnancies resulted in 12 million births, and 55% of them ended in unsafe abortions[13].” Refers to but it implies that half of the global pregnancies were unintended (so not sure what proportion are from SSA), and that of the half that were unintended, half of those ended in unsafe abortions? That seems extremely high to me, so I looked up the reference, and two things 1) Ref 13 looks incomplete in the reference list and 2) the reference said 55% resulted in abortions not that these were unsafe, but that some of these were unsafe.

• Some specificity in the intro would be helpful, only half (I think) of pregnancies are unintended, which means some are intended. Those that are intended may still threaten well-being or economic attainment, but terminology used and argument don’t make any distinction.

• Consider being very explicit at the end of the intro – its there but could be more clear – what this article was aiming to do by literally saying, “Therefore, the current study aimed to …

• “Subtheme 2: Young adolescents are aware of the negative outcomes associated with poorly managed adolescence” the phrase “poorly managed adolescence” is a bit vague I wonder about rather saying “poorly managed sexuality during adolescence”?

• Here “In order to address this vulnerability, male participants underscored the importance of providing comprehensive education on sexuality and reproductive health” you say male participants but almost all quotes are from females (which does make more sense to me)

• Re-orient the reader to the goal of the paper at the start of the discussion before diving into results.

• This sentence is a bit long and confusing: “Therefore, basing on this finding, we suggest that interventions targeting the reduction of STI and HIV transmission, along with the prevention of unintended pregnancies among adolescents, should be based on the current awareness that young individuals possess regarding the negative outcomes associated with risky sexual behaviors.”

• This is still in the manuscript: - “consisted of sexually active young individuals” I don’t think this was inclusion criteria?

7. PLOS authors have the option to publish the peer review history of their article (what does this mean?). If published, this will include your full peer review and any attached files.

Reviewer #1: **Yes: **Seyed Ali Azin

Reviewer #2: No

---

## [Author Response · Author response to Decision Letter 1]

14 Jul 2024

Comments How the comments have been addressed

Reviewer 1

The revised article provides more information and is more coherent. Mentioning details about the questions and topics of focused group discussions could help improve the article. Thank you for the feedback. We have included the interview guide used in the supplementary information (see supplementary information 1)

Reviewer 2

Thank you for this revised manuscript. Several revisions I think were helpful for clarity. I do have a few additional comments, below.

• This edit: “an unwanted pregnancy is described as a pregnancy that occurs before they have reached physical and mental maturity to fully understand the implications of their actions and provide informed consent.” With current edits, I think this sentence can just be dropped. Thank you. The sentence was deleted. (see page 4)

Paragraph 2 of the intro, there are several acronyms used that are not spelled out. Thank you for the comment. The abbreviations have been spelled out in page 4.

This section in the intro is still a bit confusing: “In 2019 alone, it was estimated that 21 million pregnancies occurred in this age group, with approximately half of them being unintended were from Sub-Saharan Africa[14].” Were approximately half unintended or half were from SSA? Thank you for the observations. Corrections were made to make the statement clearer. (see page 5 for edits made)

Similarly, I am not sure what “These unintended pregnancies resulted in 12 million births, and 55% of them ended in unsafe abortions[13].” Refers to but it implies that half of the global pregnancies were unintended (so not sure what proportion are from SSA), and that of the half that were unintended, half of those ended in unsafe abortions? That seems extremely high to me, so I looked up the reference, and two things 1) Ref 13 looks incomplete in the reference list and 2) the reference said 55% resulted in abortions not that these were unsafe, but that some of these were unsafe. Thank you for this observation. The confusion has been address and the reference was corrected. (see page 5) the reference list has been revised to make reference 13 complete.

Some specificity in the intro would be helpful, only half (I think) of pregnancies are unintended, which means some are intended. Those that are intended may still threaten well-being or economic attainment, but terminology used and argument don’t make any distinction. Thank you for the observation. With the revised version (with the previous 2 comments), this issue has been addressed.

Consider being very explicit at the end of the intro – its there but could be more clear – what this article was aiming to do by literally saying, “Therefore, the current study aimed to … Thank you for this observation. The issue raised has been addressed on page 6 of the current version.

“Subtheme 2: Young adolescents are aware of the negative outcomes associated with poorly managed adolescence” the phrase “poorly managed adolescence” is a bit vague I wonder about rather saying “poorly managed sexuality during adolescence”? Thank you for the suggestion. Comment considered. See page 14.

Here “In order to address this vulnerability, male participants underscored the importance of providing comprehensive education on sexuality and reproductive health” you say male participants but almost all quotes are from females (which does make more sense to me) Thank you for the observation. We have changed male participants to ‘participants’ as this was a common perspective by both male and female participants. See page 26.

Re-orient the reader to the goal of the paper at the start of the discussion before diving into results. Thank you for observation. The comment has been taken into consideration. See page 22

This sentence is a bit long and confusing: “Therefore, basing on this finding, we suggest that interventions targeting the reduction of STI and HIV transmission, along with the prevention of unintended pregnancies among adolescents, should be based on the current awareness that young individuals possess regarding the negative outcomes associated with risky sexual behaviors.” Thank you for this comment. We have made edits to make the sentence short and comprehensible. (see page 23)

This is still in the manuscript: - “consisted of sexually active young individuals” I don’t think this was inclusion criteria? Thank you for this observation. We have addressed this on page 23.

---

## [Editor Report · Decision Letter 2]

16 Jul 2024

Factors affecting the prevention of unwanted pregnancies among young adolescents in secondary schools in the Eastern Province of Rwanda: An explorative qualitative study

PONE-D-24-09798R2

Dear Dr. Thierry Claudien UHAWENIMANA 

We’re pleased to inform you that your manuscript has been judged scientifically suitable for publication and will be formally accepted for publication once it meets all outstanding technical requirements.

Kind regards,

Yitagesu Habtu Aweke, Ph.D

Academic Editor

PLOS ONE

---

## [Editor Report · Acceptance letter]

18 Jul 2024

PONE-D-24-09798R2 

PLOS ONE

Dear Dr. Uhawenimana, 

I'm pleased to inform you that your manuscript has been deemed suitable for publication in PLOS ONE. Congratulations! Your manuscript is now being handed over to our production team.

Kind regards, 

on behalf of

PhD Candidate Yitagesu Habtu Aweke 

Academic Editor

PLOS ONE